# Exploring Covariate and Concept Shift for Detection and Confidence Calibration of Out-of-Distribution Data

## Abstract

Moving beyond testing on in-distribution data, works on Out-of-Distribution (OOD) detection have recently increased in popularity. A recent attempt to categorize OOD data introduces the concept of near and far OOD detection. Specifically, prior works define characteristics of OOD data in terms of detection difficulty. We propose to characterize the spectrum of OOD data using two types of distribution shifts: covariate shift and concept shift, where covariate shift corresponds to change in style, e.g., noise, and concept shift indicates change in semantics. This characterization reveals that sensitivity to each type of shift is important to the detection and confidence calibration of OOD data. Consequently, we investigate score functions that capture sensitivity to each type of dataset shift and methods that improve them. To this end, we theoretically derive two score functions for OOD detection, the **covariate shift score** and **concept shift score**, based on the decomposition of KL-divergence for both scores, and propose a geometrically-inspired method (Geometric ODIN) to improve OOD detection under both shifts with only in-distribution data. Additionally, the proposed method naturally leads to an expressive post-hoc calibration function which yields state-of-the-art calibration performance on both in-distribution and out-of-distribution data. We are the first to propose a method that works well across both OOD detection and calibration, and under different types of shifts.

## 1 Introduction

Out-of-distribution (OOD) detection is a fundamental research area important for downstream tasks with open-world assumptions such as continual learning (Smith et al., 2021), open-set learning (Yu et al., 2020) and safety-critical applications such as self-driving (Bojarski et al., 2016). However, common OOD detection benchmarks[1] are not well categorized. This prevents us from more systematically studying of OOD detection. A recent attempt (Winkens et al., 2020; Fort et al., 2021) to provide a more granular characterization of OOD data introduces the concept of near OOD (Winkens et al., 2020; Fort et al., 2021). Near OOD detection is posed as a more challenging problem than far OOD detection because near OOD datasets usually share similar semantics, and style, e.g., all natural images of similar environment, as the training dataset. Existing works characterize OOD in terms of difficulty of OOD detection such as Confusion Log Probability (CLP) (Winkens et al., 2020). However, this characterization only conveys difficulty of OOD detection across a single dimension and does not expose intrinsic characteristics of OOD data.

In this paper we provide a more systematic categorization of OOD data, which motivates a more robust OOD detection method and helps us reflect on some existing approaches. Specifically, there are at least two dimensions along which we can characterize the spectrum of OOD data. Intuitively, out-of-distribution data refers to data that are sampled from distributions different from the training distribution. This is well known in machine learning as *distribution shift* (Moreno-Torres et al., 2012). There are two dominant shift types: *covariate shift*[2] and *concept shift*. The former usually

---

[1]Texturess (Cimpoi et al., 2014), SVHN (Netzer et al., 2011), Place365 (Zhou et al., 2017), LSUN-Crop/Resize (Yu et al., 2015), and iSUN (Xu et al., 2015)

[2]Covariate shift is often associated with model calibration (Ovadia et al., 2019; Chan et al., 2020)

refers to change in style, e.g., clean to noised and natural images to cartoon, and the latter refers to change in semantics, e.g., *dog* to *cat*. Even though existing works Hsu et al. (2020) have acknowledged the different types of OOD datasets and the difficulty of detecting them, their methods do not distinguish them and do not explicitly disentangle them. To study these two distribution shifts, we propose to create a benchmark with multiple magnitude of distribution shift in each dimension. Specifically, we use corrupted CIFAR10C/CIFAR100C (Hendrycks & Dietterich, 2019), originally developed for testing robustness, with varying degrees of severity to represent increasing covariate shift; we also introduce a new benchmark, CIFAR100 Splits, which divides the CIFAR100 dataset into 10 splits with increasing conceptual shift from CIFAR10 classes, measured by semantic similarity in a word embedding space (Pennington et al., 2014). We show that the increasing conceptual difference measured by semantic similarity in language translates to a spectrum of OOD datasets under gradual concept shift measured by the difficulty of OOD detection in the image space.

To address these two distribution shifts, we need to approach it from two aspects: *representation* and *modeling*, analogous to the role of entropy and Bayesian inference in uncertainty quantification (Depeweg et al., 2018) where entropy is a score function that represents uncertainty and Bayesian inference is a method that models uncertainty (Hüllermeier & Waegeman, 2021). To **represent** these two shifts, we can derive score functions which output scalars indicating the severity of distribution shift. Specifically we derive two score functions by expanding and decomposing the KL divergence between a predictive distribution from a classifier and a discrete uniform distribution. Arising from the decomposition is a *covariate shift score* which is a function of feature norms and a *concept shift score* which is a function of feature angles. To **model** these two shifts, and motivated by the covariate score (a function of norms) and the concept score (a function of angles), we propose to directly improve the sensitivity of norms and angles to distribution shift through a geometric perspective (Tian et al., 2021). Specifically, we adopt the Geometric Sensitivity Decomposition proposed in Tian et al. (2021), originally developed for model calibration, which decomposes norms and angles into a variance component and a scalar offset, and improve it by parametrizng the scalar offsets as standalone networks. Our approach improves detection of OOD data under both covariate and concept shifts.

While OOD detection and calibration[3] has been studied separately in the literature, they share the same underlying motivation: confidence/uncertainty should be low/high when distribution shift occurs. Our method to improve OOD detection naturally yields a powerful calibration function in the family of intra class-preserving functions (Rahimi et al., 2020). This class of functions provides enough expressive power to calibrate complex decision boundaries in neural networks and proves to be effective in calibration on in-distribution data. Therefore, we apply a post-calibration (Guo et al., 2017) method on a validation set as described in those works. Additionally, thanks to the improved sensitivity to distribution shift, our model also is on par with state-of-the-art calibration performance on distribution shifted data as well. To summarize our contributions:

- We propose to characterize the spectrum of OOD data in terms of covariate and concept shift and unify the notion of near OOD in different literature (Sec. 3.1). Additionally, we construct a new benchmark (e.g., CIFAR100 Splits) to represent gradual distribution shift in each direction (Sec. 4.1.2 and 4.1.3).
- To represent distribution shifts, we analytically derive two OOD score functions based on KL-divergence to capture both shifts more effectively (Sec. 3.2).
- To model distribution shifts, we propose Geometric ODIN to improve the sensitivity of neural networks during training (Sec. 3.3) and calibration during inference (Sec. 3.4). This method achieves state-of-the-art performance on both detection (Sec. 4.1) and calibration (Sec. 4.2) of OOD data.

## 2 RELATED WORK

**Out-of-Distribution (OOD) detection** methods can be largely divided into two camps depending on whether they require OOD data during training. Hendrycks et al. (2018); Thulasidasan et al. (2020); Roy et al. (2021) leverages anomalous data in training. Our method belongs to the class of methods that do not assume the availability of OOD data during training. Hendrycks & Gimpel (2016) uses the maximum softmax probability (MSP) to detect incorrect predictions and OOD data. Lee et al.

---

[3]OOD detection focuses on concept-shifted data while calibration focuses on covariate-shifted data.

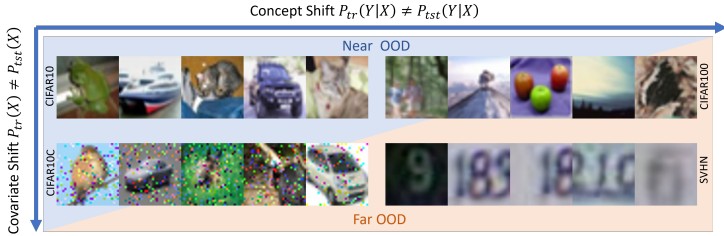

Figure 1: Illustration of near and far Out-of-Distribution data. Top left: CIFAR10 dataset (training). Top right: CIFAR100 dataset. Lower left: Corrupted CIFAR10. Lower right: SVHN dataset.

(2018) proposes to use Mahalanobis distance by fitting a Gaussian mixture model (GMM) in the feature space. Mukhoti et al. (2021) uses log density of the GMM model instead. Liu et al. (2020b) uses an energy score as the uncertainty metric. ODIN (Liang et al., 2017) uses a combination of input processing and post-training tuning to improve OOD detection performance. Generalized ODIN Hsu et al. (2020) (also Techapanurak et al. (2019)) includes an additional network in the last layer to improve OOD detection during training. There are other interesting OOD detection approaches without OOD data such as using contrastive learning with various transformations (Winkens et al., 2020; Tack et al., 2020), training a deep ensemble of multiple models (Lakshminarayanan et al., 2016) and leveraging large pretrained models (Fort et al., 2021). They require extended training time, hyperparameter tuning and careful selections of transformations, whereas our method does not introduce any hyperparameters and has negligible influence on standard cross-entropy training time.

**Model Calibration** methods can also be largely divided in two categories: 1) training time calibration using augmentations (Thulasidasan et al., 2019; Jang et al., 2021), using modified losses (Kumar et al., 2018); 2) post-hoc calibration (Guo et al., 2017; Kull et al., 2019; Kumar et al., 2019; Rahimi et al., 2020). Recently, Rahimi et al. (2020) formally generalizes a family of expressive functions for calibration, *the intra order-preserving functions*. This class of functions has more representation power to calibrate more complex decision boundaries in neural networks. Our proposed method belongs to this class of functions. However, all these works focus on calibration of in-distribution data. To obtain better calibration on OOD data, on which the confidence of a model needs to decrease accordingly, sensitivity and deterministic uncertainty modeling is explored by SNGP (Van Amersfoort et al., 2020) and DUQ (Liu et al., 2020a). Our proposed model not only belongs to the family of intra order-preserving functions, which ensures good in-distribution calibration, but also improves sensitivity to distribution shift, which improves out-of-distribution calibration simultaneously. Please refer to Appendix 6.8 for a more detailed discussion on related works.

## 3  METHOD

### 3.1  MOTIVATION: COVARIATE AND CONCEPT SHIFT

We follow Moreno-Torres et al. (2012) for the formal definition of distribution shift, covariate shift and concept shift. Let $X \in \mathcal{R}^D$ denote the covariate which is the input and $Y \in \mathcal{R}$ denote the output label. *Distribution shift* happens when the training joint distribution is not equal to the testing joint distribution $P_{tr}(X, Y) \neq P_{tst}(X, Y)$. *Covariate shift* appears when $P_{tr}(Y|X) = P_{tst}(Y|X)$ and $P_{tr}(X) \neq P_{tst}(X)$. *Concept shift* appears when $P_{tr}(Y|X) \neq P_{tst}(Y|X)$ and $P_{tr}(X) = P_{tst}(X)$. However, in the image domain, it is rare that such concept shift occurs without changes in $P(X)$. We therefore modify the equality in concept shift to $P_{tr}(X) \approx P_{tst}(X)$ where superficial, low-level statistics may be retained.

Specifically, covariate shift happens when the testing data is non-semantically different from the training data and concept shift happens when the testing data is semantically different from the training data. We illustrate a spectrum of OOD dataset in Fig. 1. In this example, CIFAR10 is the training dataset. CIFAR100 represents concept shift because CIFAR100 has a non-overlapping label space, i.e., $P_{tr}(Y|X) \neq P_{tst}(Y|X)$, but similar style with CIFAR10. The corrupted CIFAR10C dataset (Hendrycks & Dietterich, 2019) represents covariate shift because it has the same labels but

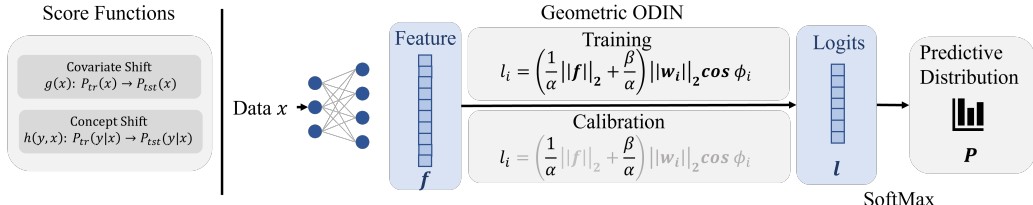

Figure 2: **Diagram of Score functions, Geometric ODIN training and calibration** The paper proposes two score functions and an OOD detection and calibration method: Geometric ODIN. Loss is backprobagated to all components during training and only to $\alpha$ and $\beta$ during calibration.

different style, i.e., $P_{tr}(X) \neq P_{tst}(X)$, compared to CIFAR10. SVHN represents both covariate and concept shift due to its non-overlapping label space and very different style.

As shown in Fig. 2, the first goal of this paper is to derive score functions to represent the shift in either $P(X)$ or $P(Y|X)$. We denote the score function that reflects change from $P_{tr}(X)$ to $P_{tst}(X)$ as the **covariate shift score function**: $g(x) : \mathcal{R}^D \to \mathcal{R}$ and the score function that reflects change from $P_{tr}(Y|X)$ to $P_{tst}(Y|X)$ as the **concept shift score function**: $h(y, x) : \mathcal{Y} \times \mathcal{R}^D \to \mathcal{R}$. The second goal is to improve the sensitivity of these scores to their corresponding distribution shift. Specifically, we propose a geometrically inspired **o**ut-of-distribution **d**etection method with only **in**-distribution data (**Geometric ODIN**). The third goal is to show that, unlike prior works which treat OOD detection (Mukhoti et al., 2021) and calibration (Van Amersfoort et al., 2020; Liu et al., 2020a) separately, our proposed Geomeric ODIN naturally leads to an expressive calibration function in the family of *intra order-preserving functions* (Rahimi et al., 2020), which ensures good calibration.

## 3.2 COVARIATE AND CONCEPT SCORE FUNCTIONS

In this section, we theoretically derive two score functions, $g(x)$ and $h(y, x)$, based on the KL-divergence between a uniform distribution $\mathbf{U}$ and a predicted distribution $\mathbf{P} \in \mathcal{R}^M$, where $M$ is the number of classes. By starting from KL-divergence, we hinge the subsequent derivation of score functions on a physical meaningful uncertainty measure, i.e., how far the predicted distribution is from a uniform distribution. This relationship ensures a natural interpretation of score functions because predictions on distribution shifted data should have larger uncertainty, i.e. smaller distance from uniform. We are specifically interested in softmax-linear models for classification. They typically consist of a feature extractor and a linear layer followed by a softmax activation. Let $\mathbf{f} \in \mathcal{R}^D$ denote a feature vector from the feature extractor[4]. The output of the linear layer, i.e., logits, $\mathbf{l} \in \mathcal{R}^M = <\mathbf{f}, \mathbf{W}>$ is defined as the inner product between the feature vector and a weight matrix in the linear layer. Let $l_i = \|\mathbf{f}\|_2 \|\mathbf{w}_i\|_2 \cos \phi_i$ denote the $i$th logit, $P_i = \frac{\exp l_i}{\sum_{j=1}^M \exp l_j}$ denote the predicted probability of the $i$th class. The KL-divergence $\mathbb{KL}(\mathbf{U}\|\mathbf{P})$ can be written as following:

$$\mathbb{KL}(\mathbf{U}\|\mathbf{P}) = -\sum_{i=1}^M \frac{1}{M} \ln M P_i = \underbrace{\ln \sum_{j=1}^M \exp l_j}_{Log-Sum-Exp} - \frac{1}{M}\sum_{i=1}^M l_i - \ln M \qquad (1)$$

Now we can use the inequality property of Log-Sum-Exp (LSE)[5] functions in Eq. 2 to bound Eq. 1.

$$\max_j l_j \leq \ln \sum_{j=1}^M \exp l_j \leq \max_j l_j + \ln M \qquad (2)$$

Therefore the KL-divergence (Eq. 1) can be bounded as follows:

$$\mathcal{U} - \ln M \leq \mathbb{KL}(\mathbf{U}\|\mathbf{P}) \leq \mathcal{U} \qquad (3)$$

---

[4]Bold letter indicates vectors

[5]Note that the negative LSE function is also defined as *free energy* in Liu et al. (2020b).

where $\mathcal{U} = \max_j l_j - \frac{1}{M} \sum_{i=1}^{M} l_i$. $\mathcal{U}$ can be further decomposed into two multiplicative components by plugging in the definition of logits $l_i$:

$$\mathcal{U} = \max_j l_j - \frac{1}{M} \sum_{i=1}^{M} l_i = \overbrace{\|\mathbf{f}\|_2}^{g(x)} \underbrace{\left( \max_j \|\mathbf{w}_j\|_2 \cos \phi_j - \frac{1}{M} \sum_{i=1}^{M} \|\mathbf{w}_i\|_2 \cos \phi_i \right)}_{h(y,x)} \tag{4}$$

We define the **covariate shift score function** as $g(x) \triangleq \|\mathbf{f}\|_2$ because the norm of a feature vector is the sum of squared activation values and only depends on the input. Intuitively, activation of a neural network on covariate-shifted data should be weaker than in-distribution data. Therefore, $g(x)$ assigns a higher value to in-distribution data than to OOD data. We define the **concept shift score function** as $h(y,x) \triangleq \max_j \|\mathbf{w}_j\|_2 \cos \phi_j - \frac{1}{M} \sum_{i=1}^{M} \|\mathbf{w}_i\|_2 \cos \phi_i$ because it is the difference between the cosine distance of the *predicted* class and the average cosine distance of *all* classes and depends on both the input and final class membership, assigned by the max operator. Intuitively, class assignment should be less obvious under concept shift and the difference should be small. Consequently, $h(y,x)$ assigns a higher value to in-distribution data than to OOD data. In retrospect, our definition of covariate shift and concept shift scores supports existing findings that the feature norms correspond to intra-class variance and angles reflect inter-class variation (Liu et al., 2018). Intuitively, covariate shift represents non-semantic change within a specific class, i.e., intra-class variance; concept shift represents semantic changes, i.e, inter-class variation. Here we formalize the intuition and observations in Liu et al. (2018) as score functions derived analytically from a KL-divergence viewpoint. We provide an extended discussion of these scores in Appendix 7.

More importantly, the **combined score function** $\mathcal{U}$ (Eq. 4) carries a physical meaning: it bounds the KL-divergence between a uniform distribution and the predictive distribution. Intuitively, a small $\mathcal{U}$ indicates large uncertainty because $\mathcal{U}$ upper bounds $\mathbb{KL}(\mathbf{U}||\mathbf{P})$, and a large $\mathcal{U}$ indicates small uncertainty because it also appears in the lower bound. We can derive $\mathcal{U}$ also by Taylor-expanding the softmax equation as in Liang et al. (2017), which claims that $\mathcal{U}$ is responsible for good OOD detection. Our findings confirm this and push it further by decomposing it into different components. A similar scoring function, $S(x) = \|\mathbf{f}\|_2 \max_j \|\mathbf{w}_j\|_2 \cos \phi_j$, is used in Tack et al. (2020), but is only empirically motivated based on observations with limited analytical insights such as its relationship to uncertainty and the functionality of each of components. In contrast, our derivation clearly shows the relationship between these score functions and the KL-divergence, which is as an uncertainty measure, and disentangles their roles. We will compare the $g(x)$, $h(y,x)$ and $\mathcal{U}$ as score functions in subsequent experiments and analyze their respective sensitivity to different shifts (Sec. 4.1).

## 3.3 GEOMETRIC OUT-OF-DISTRIBUTION DETECTION WITH IN-DISTRIBUTION DATA

As derived in Eq. 4, the covariate score is a function of feature norms and the concept score is a function of feature angles. Consequently, improving the sensitivity of feature norms and feature angles to data shifts seems to be the natural next step to improve OOD detection. Therefore, we adopt Geometric Sensitivity Decomposition (GSD) (Tian et al., 2021) (reviewed in appendix 6.3) to improve sensitivity to covariate and concept shifts. Specifically, GSD improves sensitivity by extracting sensitive components from norms $\|\mathbf{f}^*\|_2$[6] and angles $|\phi_i^*|$ through a decomposition of them into: a scalar offset and a variance component as shown in Eq. 5. Scalar offsets $\mathcal{C}_f$ and $\mathcal{C}_\phi$ minimize the loss on the training set and the variance components $\mathbf{f}$ and $\phi_i$ account *sensitively* for variances in samples.

$$\|\mathbf{f}^*\|_2 = \|\mathbf{f}\|_2 + \mathcal{C}_f, \quad |\phi_i^*| = |\phi_i| - |\mathcal{C}_\phi| \tag{5}$$

With the decomposed components, the *original* logit $l_i^* = \|\mathbf{f}^*\|_2 \|\mathbf{w_i}\|_2 \cos \phi_i^*$, can be written as:

$$l_i^* = \|\mathbf{f}^*\|_2 \|\mathbf{w_i}\|_2 \cos \phi_i^* \approx l_i = \left( \underbrace{\frac{1}{\cos \mathcal{C}_\phi}}_{\alpha} \|\mathbf{f}\|_2 + \underbrace{\frac{1}{\cos \mathcal{C}_\phi}}_{\alpha} \overbrace{\mathcal{C}_f}^{\beta} \right) \|\mathbf{w_i}\|_2 \cos \phi_i \tag{6}$$

---

[6]The superscript $*$ denotes the *original* component before decomposition.

where $l_i$ denote the *new* $i$th logit. In Eq. 6, the *new*[7] feature $\mathbf{f}$ is a direct output of a feature extractor, and is modified by $\alpha$ and $\beta$ as also illustrated Fig. 2. Note that the calculation of score functions in Sec. 3.2 only uses the feature and is independent of $\alpha$ and $\beta$.

Because $\cos \mathcal{C}_\phi$ and $\mathcal{C}_f$ are scalar offsets, we can parametrize them separately from the main network. Unlike GSD which parametrizes them as *instance-independent* scalars, inspired by Hsu et al. (2020); Techapanurak et al. (2019), we make $\alpha(\mathbf{f})$ and $\beta(\mathbf{f})$ instance-dependent scalars and use a single linear layer to learn them. To enforce numerical constraints, i.e., $0 < \alpha < 1$ and $\beta > 0$, $\alpha(\mathbf{f})$ uses a *sigmoid activation* and $\beta(\mathbf{f})$ uses a *softplus activation*. Finally, the relaxed output is:

$$P(Y = i | x) = \frac{\exp l_i}{\sum_{j=1}^{M} \exp l_j} = \frac{\exp \left( \left( \frac{1}{\alpha(\mathbf{f})} \|\mathbf{f}\|_2 + \frac{\beta(\mathbf{f})}{\alpha(\mathbf{f})} \right) \|\mathbf{w_i}\|_2 \cos \phi_i \right)}{\sum_{j=1}^{M} \exp \left( \left( \frac{1}{\alpha(\mathbf{f})} \|\mathbf{f}\|_2 + \frac{\beta(\mathbf{f})}{\alpha(\mathbf{f})} \right) \|\mathbf{w_j}\|_2 \cos \phi_j \right)} \tag{7}$$

Now the *new* predicted norm $\|\mathbf{f}\|_2$ and angle $\phi_i$ are more sensitive to input changes because they encode variances in samples as shown in Eq. 5. Therefore, including $\beta$ (related to norms) improves sensitivity to covariate shift and including $\alpha$ (related to angles) improves sensitivity to concept shift. Note that, under this construction, Generalized ODIN (Hsu et al., 2020) is a special case of our proposed method. Generalized ODIN only includes the $\alpha(\mathbf{f})$ which only improves angle sensitivity but not norm sensitivity. Unlike Hsu et al. (2020)'s probabilistic perspective[8], our model builds on a geometric perspective and captures both covariate and concept shifts by improving norm and angle sensitivity. The new model can be trained identically as the vanilla network without additional hyperparameter tuning and extended training time. Combined with score functions derived in Sec. 3.2, Geometric ODIN achieves state-of-the-art OOD detection performance (Sec. 4.1)

### 3.4 CALIBRATION OF GEOMETRIC ODIN

Like most softmax-linear classification models, Geometric ODIN also suffers from miscalibration right after training. Specifically, predictions tend to be overconfident (Guo et al., 2017). In the case of Geometric ODIN, the cause of overconfidence is obvious by construction. The offset scalars $\beta(\mathbf{f})$ and $\alpha(\mathbf{f})$ are designed to minimize the training loss. Practically, they are optimized to push the predicted probability of the ground truth class to be close to 1 during training, and this intended behavior unsolicitedly continues to inference time when accuracy is not as high as during training. Nevertheless, our construction of $\alpha(\mathbf{f})$ and $\beta(\mathbf{f})$ belongs to a family of *intra order-preserving* functions, which has potentially strong representation to calibrate complex functions in deep networks according to Theorem 1 in Rahimi et al. (2020). To prove that, it's suffice to show that that $\alpha(\mathbf{l})$ and $\beta(\mathbf{l})$ [9] forms a strictly positive function, $m(\mathbf{l})$ as shown in Eq. 8

$$\bar{l}_i = \left( \frac{1}{\alpha(\mathbf{l})} \|\mathbf{f}\|_2 + \frac{\beta(\mathbf{l})}{\alpha(\mathbf{l})} \right) \|\mathbf{w_i}\|_2 \cos \phi_i = \underbrace{\left( \frac{1}{\alpha(\mathbf{l})} + \frac{\beta(\mathbf{l})}{\alpha(\mathbf{l})\|\mathbf{f}\|_2} \right)}_{m(\mathbf{l})} \overbrace{\|\mathbf{f}\|_2 \|\mathbf{w_i}\|_2 \cos \phi_i}^{l_i} \tag{8}$$

Because $0 < \alpha(\mathbf{l}) < 1$ and $\beta(\mathbf{l}) > 0$, the function $m(\mathbf{l})$ is strictly positive and thus satisfies Theorem 1 in Rahimi et al. (2020) (See appendix 6.4 for more details). We follow a simple procedure proposed in Guo et al. (2017); Rahimi et al. (2020) to calibrate Geometric ODIN by minimizing the negative log likelihood (NLL) on a evaluation dataset for a few epochs. During calibration the gradients to other parts of the model are stopped and only these two linear layers, $\alpha(\mathbf{l})$ and $\beta(\mathbf{l})$, are optimized (Fig. 2). Naturally with the improved sensitivity and calibrated functions, the calibration of Geometric ODIN is on par with state-of-the-art models on both in-distribution and OOD data. Note that, as mentioned in Sec. 3.3, because the calculation of score functions for OOD detection only depends on the feature $\mathbf{f}$ and is independent of $\alpha$ and $\beta$, calibration does not affect the OOD detection performance and does not change prediction results due to the order-preserving property.

---

[7] Even though both $\mathbf{f}^*$ and $\mathbf{f}$ are outputs directly from the feature extractor, we use *original* and *new* to indicate whether GSD is applied.

[8] $\alpha(\mathbf{f})$ is interpreted as $P(d_{in}|x)$, the probability of $x$ being in-distribution.

[9] Following Rahimi et al. (2020), which uses logits $\mathbf{l}$ as input instead of the feature $\mathbf{f}$. Our derivation in previous sections is still valid because $\mathbf{l}$ solely depends on $\mathbf{f}$ given a model.

| AUROC↑ | Score Functions | ID:CIFAR100 | | | ID:CIFAR10 | | | |
|---|---|---|---|---|---|---|---|---|
| | | Near CIFAR10 | Far SVHN | Far TIN(R) | Near CIFAR100 | Near CIFAR10C | Far SVHN | Far TIN(R) |
| Vanilla Wide-ResNet-28-10 (Zagoruyko & Komodakis, 2016) | MSP (Hendrycks & Gimpel, 2016) | 80.68±0.34 | 77.37±2.25 | 81.65±0.14 | 88.93±0.37 | 70.58±0.59 | 93.66±1.79 | 87.98±0.35 |
| | Energy (Liu et al., 2020b) | **80.74±0.45** | 79.48±2.91 | 82.04±0.14 | 88.84±0.44 | 70.60±0.52 | 94.39±2.30 | 88.16±0.39 |
| | Mahanobis (Lee et al., 2018) | 66.72±0.77 | 93.55±1.18 | 76.54±0.31 | 87.26±1.21 | 68.38±0.50 | 99.19±0.22 | 87.33±1.39 |
| | GMM Density (Mukhoti et al., 2021) | 66.75±0.74 | 93.91±0.71 | 76.58±0.29 | 87.26±1.20 | 75.71±0.80 | 99.19±0.22 | 87.33±1.39 |
| DUQ (Van Amersfoort et al., 2020) | Kernel Distance | – | – | – | 85.92±0.35* | – | 93.71±0.61* | – |
| SNGP (Liu et al., 2020a) | SoftMax Entropy | – | 85.71±0.81* | – | 91.13±0.15* | – | 94.0±1.30* | – |
| DDU (Mukhoti et al., 2021) | GMM Density | 67.65±0.20 | 92.59±1.47 | 77.72±0.15 | 90.69±0.42 | 76.00±0.00 | 97.12±1.21 | 84.89±1.01 |
| Hyper-Free/Generalized ODIN | Cosine Similarity | 76.90±0.30 | 95.39±1.31 | 82.93±0.18 | 92.28±0.16 | 75.84±0.70 | **99.56±0.12** | 92.03±0.15 |
| 5-Ensemble (Lakshminarayanan et al., 2016) | SoftMax Entropy | – | 79.54±0.91* | – | 92.13±0.02* | – | 97.73±0.31* | – |
| Ours: α(x)-only | h(y, x) | 79.24±0.37 | 83.75±3.30 | 82.67±0.22 | 92.28±0.15 | 77.00±0.00 | 97.56±0.90 | 91.83±0.09 |
| Ours: α-β | h(y, x) | 79.72±0.41 | 88.72±2.99 | 82.89±0.24 | 89.08±0.24 | 73.80±0.45 | 98.42±0.21 | 90.87±0.10 |
| | g(x) | 60.70±0.56 | 93.15±2.06 | 72.34±0.42 | 89.08±0.24 | 76.80±1.10 | 99.40±0.14 | 90.79±0.39 |
| | $\mathcal{U}$ | 71.42±0.27 | **95.77±0.33** | 80.86±0.53 | **92.31±0.21** | **78.00±0.71** | 99.54±0.08 | **92.85±0.19** |

Table 1: **AUROC ↑ for Near and Far OOD detection**. Results are averaged over 5 runs. Our $\alpha(x)$-only model is the same as Generalized ODIN (Hsu et al., 2020) ($h^I(x)$ variant) without its input processing. Hyper-free (Techapanurak et al., 2019) and Generalized ODIN ($h^C(x)$ variant) are the same. * denotes results from Mukhoti et al. (2021).

## 4 EXPERIMENTS

In this section, we present results for Out-of-Distribution detection in Sec. 4.1 and calibration in Sec. 4.2. We are the first to present a method that works well on both OOD detection and calibration, disentangling shift types. **Implementation:** Following prior works (Liu et al., 2020a; Van Amersfoort et al., 2020; Mukhoti et al., 2021), we use Wide-ResNet-28-10 (Mukhoti et al., 2021) for all experiments. We train the model using SGD with an initial learning rate of 0.1 for 200 epochs. The learning rate is annealed with a cosine scheduler (Loshchilov & Hutter, 2016) (more details in appendix 6.5). **Metrics:** For OOD detection in Sec. 4.1, we use the common AUROC and TNR@TPR95 as benchmark metrics. For calibration in Sec. 4.2, we use the Expected Calibration Error (ECE) (Guo et al., 2017) with 15 bins and Negative Log Likelihood (NLL) which is a strictly proper scoring rule (Gneiting & Raftery, 2007). **Datasets:** CIFAR10 (Krizhevsky et al., a) and CIFAR100 (Krizhevsky et al., b) are considered near OOD datasets (Winkens et al., 2020; Fort et al., 2021) to each other. SVHN (Netzer et al., 2011) is considered a far OOD dataset to both CIFAR10 and CIFAR100 due to its shift in both concept and style. The CIFAR10C/CIFAR100C (Hendrycks & Dietterich, 2019) dataset is a variant of CIFAR10/CIFAR100 corrupted by 15 types of noises with 5 degrees of severity. We also introduce CIRAF100 Splits benchmark which consists of 10 datasets with increasing concept shift from CIFAR10 classes (Sec. 4.1.3).

### 4.1 OUT-OF-DISTRIBUTION DETECTION RESULTS

### 4.1.1 NEAR AND FAR OOD DETECTION

**The $\alpha$-$\beta$ model is the best.** In Tab. 1, we present OOD detection results against state-of-the-art methods on existing near and far OOD categorization. For near OOD detection under strong concept shift, CIFAR10 (ID) vs. CIFAR100 (OOD), both our $\alpha$-only and $\alpha$-$\beta$ variants achieve the the best performance. This demonstrates that $\alpha(x)$ improves the sensitivity of angles and hence the sensitivity to concept-shifted data. For near OOD detection under strong covariate shift, CIFAR10 (ID) vs. CIFAR10C (OOD), the $\alpha$-$\beta$ variant achieves the the best performance. This suggests that $\beta(x)$ improves the sensitivity of norms and hence the sensitivity to covariate-shifted data. In CIFAR100 (ID) vs. CIFAR10 (OOD) experiments, the performance of the $\alpha$-only and $\alpha$-$\beta$ variants are within variance and is close to some other compared methods, we can not make clear observations from those experiments[10]. For far OOD detection, CIFAR10/CIFAR100 (ID) vs. SVHN (OOD), the $\alpha$-$\beta$ model achieves state-of-the-art performance. This reconfirms that $\beta(x)$ improves sensitivity to covariate-shifted data, because SVHN has both covariate and concept shifts compared to the CIFAR datasets, and the $\alpha$-$\beta$ model outperforms the $\alpha$-only variant, which only improves on concept shift, by a noticeable margin. In terms of score functions, the best performing one for the $\alpha$-$\beta$ model is $\mathcal{U}$, which is a product of $g(x)$ and $h(y, x)$ (Sec. 3.2), while that of the $\alpha$-only model is $h(y, x)$. This shows that depending on which component is more sensitive, different scoring functions are

---

[10]Other confounding factors could contribute to the close performance. Prior works either omit comparisons under these settings (Mukhoti et al., 2021) or report only marginal improvement (Winkens et al., 2020)

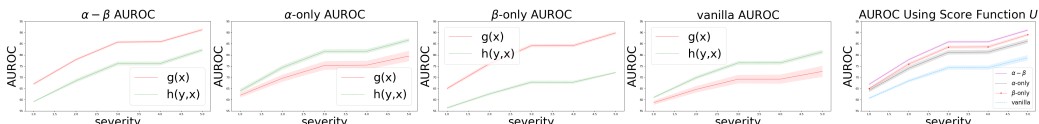

a $\alpha$-$\beta$, $\alpha$-**only**, $\beta$-**only and vanilla models using score functions** $g(x)$ **and** $h(y,x)$     b **All models using** $\mathcal{U}$

Figure 3: **Capturing Covariate Shift (Motion Blur)** All results are averaged over 5 runs. Modeling covairate shift ($\alpha$-$\beta$ model) yields the best performance as shown in Fig. 3b. $g(x)$ is more responsive to covariate shift than $h(y,x)$ as shown in Fig. 3a with the $\alpha$-$\beta$ model.

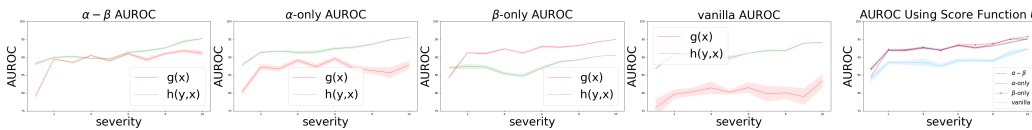

a $\alpha$-$\beta$, $\alpha$-**only**, $\beta$-**only and vanilla models using score functions** $g(x)$ **and** $h(y,x)$     b **All models using** $\mathcal{U}$

Figure 4: **Capturing Concept Shift (CIFAR100 Splits)** All results are averaged over 5 runs. Modeling concept shift (both $\alpha$-only and $\alpha$-$\beta$ models) yields the best performance as shown in Fig. 4b. $h(y,x)$ is more responsive to concept shift than $g(x)$ as shown in Fig. 4a with the $\alpha$-$\beta$ model.

preferred. When the sensitivity of both norms and angles are improved, as in the $\alpha$-$\beta$ variant, the combined score function $\mathcal{U}$ performs well under different distribution shifts.

### 4.1.2 OOD Detection under Covariate Shift with Image Corruption

**The** $g(x)$ **score captures covariate shift.** We compare the $\alpha$-$\beta$, $\alpha$-only ($\beta = 0$), $\beta$-only ($\alpha = 1$) variants and the vanilla model on CIFAR10C (Hendrycks & Dietterich, 2019) corrupted by motion blur in Fig. 3 with increasing degrees of noise (see appendix 6.7 for additional experiments). From 3a, we observe that 1) as covariate shift severity increases, OOD detection becomes easier because AUROC increases with increasing severity. 2) the vanilla model is more sensitive to the concept shift component because $h(y,x) > g(x)$ in the vanilla model plot even though covariate shift is the dominant distribution shift in this example. 3) when sensitivity to both covariate and concept shift is improved , the $\alpha$-$\beta$ model becomes more sensitive to the covariate shift component. This suggests that the dominant shift in this example is indeed covariate shift. From Fig. 3b, we observe that the $\alpha$-$\beta$ model outperforms the $\beta$-only model using the combined score function $\mathcal{U}$. This suggests that improving sensitivity to both shifts and using $\mathcal{U}$ yield the best OOD detection performance. In retrospect, the performance of OOD detection has always relied on two components: the model and the score function. Some works propose more responsive score functions, e.g., energy in Liu et al. (2020b), and some works propose more sensitive models, e.g., DDU in Mukhoti et al. (2021) to improve the performance of existing score functions.

### 4.1.3 OOD Detection under Concept Shift with CIFAR100 Special Splits

Finding a dataset to benchmark gradual concept shift is not straightforward because concept shift is traditionally thought as binary: overlapping or non-overlapping. However, not all non-overlapping labels are the same. For example, *pickup truck* (CIFAR100) is semantically much closer to *truck* (CIFAR10) than *sunflowers* (CIFAR100) is. To create this gradual concept/semantic shift, we propose to divide the CIFAR100 dataset into 10 sub-datasets with increasing conceptual difference from CIFAR10 classes. Specially, we use 300 dimensional Glove word embeddings[11] (Pennington et al., 2014) trained on the entire wikipedia2014 and Gigaword5 (Napoles et al., 2012) to measure semantic closeness (inner product) between CIFAR100 and CIFAR10 classes. The result is 10 subdatasets split from CIFAR100. Please refer to appendix 6.6 for the full splits.

**The** $h(y,x)$ **score captures concept shift.** Following the previous section, we benchmark the $\alpha$-$\beta$, $\alpha$-only, $\beta$-only variants and the vanilla model on the newly created CIFAR100 Splits. From Fig. 4a, we observe that 1) as concept shift severity increases, OOD detection becomes easier because AU-

---

[11] https://nlp.stanford.edu/projects/glove/

ROC increases with increasing severity. 2) both concept shift and covariate shift are present because AUROC using either $h(y, x)$ or $g(x)$ increases. 3) the vanilla model is dominantly more sensitive to the concept shift component because concept shift is the dominant distribution shift in CIFAR100 Splits by construction and vanilla ResNet is more sensitive to concept shift (the same behavior is also observed on covariate-shift-heavy data in Sec. 4.1.2). 4) when sensitivity to both shifts is improved, the $\alpha$-$\beta$ model is still more sensitive to concept shift ($h(y, x) > g(x)$). This reconfirms that the dominant shift type is indeed concept shift. From Fig. 4b, interestingly, we observe that all three variants perform similarly and all outperform the vanilla model. Combined with previous observations that the dataset has strong concept shift and the vanilla model is already very sensitive to concept shift, improving sensitivity to the covariate shift component yields equally good performance as improving sensitivity to both shifts.

## 4.2 Out-of-Distribution Calibration Results

| | Accuracy ↑ | | ECE ↓ | | NLL ↓ | |
|---|---|---|---|---|---|---|
| | Clean | Corrupted | Clean | Corrupted | Clean | Corrupted |
| Vanilla | **96.23**±**0.13** | 69.78±1.22 | 0.015±0.001 | 0.148±0.008 | 0.148±0.005 | 1.107±0.042 |
| Temp Scaling (Guo et al., 2017) | 96.23±0.13 | 69.78±1.22 | 0.003±0.001 | 0.107±0.009 | 0.131±0.003 | 0.906±0.029 |
| Matrix scaling (Guo et al., 2017) | 95.98±0.10 | 69.81±1.11 | 0.005±0.000 | 0.107±0.009 | 0.145±0.008 | 0.966±0.041 |
| Dirichlet (Kull et al., 2019) | 96.10±0.10 | 69.75±1.09 | 0.004±0.002 | 0.114±0.007 | 0.130±0.004 | 0.977±0.032 |
| DUQ (Van Amersfoort et al., 2020)† | 94.7±0.02 | 71.6±0.02 | 0.034±0.002 | 0.183±0.011 | 0.239±0.02 | 1.348±0.01 |
| SNGP (Liu et al., 2020a)† | 95.9±0.01 | 74.6±0.01 | 0.018±0.001 | 0.090±0.012 | 0.138±0.01 | 0.935±0.01 |
| GSD (Tian et al., 2021) | 95.9±0.01 | **74.9**±**0.05** | 0.008±0.002 | 0.085±0.012 | 0.140±0.004 | **0.853**±**0.039** |
| Ours: $\alpha$-$\beta$ | 95.99±0.12 | 70.41±0.55 | **0.001**±**0.000** | **0.071**±**0.0112** | **0.130**±**0.003** | 0.854±0.029 |

Table 2: **Calibration on CIFAR10** averaged over 5 seed. † denotes results from Liu et al. (2020a).

| | Accuracy ↑ | | ECE ↓ | | NLL ↓ | |
|---|---|---|---|---|---|---|
| | Clean | Corrupted | Clean | Corrupted | Clean | Corrupted |
| Vanilla | **80.66**± **0.20** | **50.25**±**0.48** | 0.035±0.002 | 0.171±0.017 | 0.774±0.007 | 2.384±0.039 |
| Temp Scaling (Guo et al., 2017) | 80.99±0.20 | 50.25±0.48 | 0.033±0.002 | 0.163±0.017 | 0.776±0.006 | 2.368±0.039 |
| Matrix scaling (Guo et al., 2017) | 79.31 ±0.25 | 48.74±0.55 | 0.03±0.003 | 0.160±0.016 | 0.791±0.013 | 2.532±0.051 |
| Dirichlet (Kull et al., 2019) | 80.70±0.36 | 50.06±0.46 | 0.015±0.002 | 0.144±0.017 | **0.743**±**0.025** | 2.346±0.017 |
| DUQ (Van Amersfoort et al., 2020)† | 78.5±0.02 | 50.4±0.02 | 0.119±0.001 | 0.281±0.012 | 0.980±0.02 | 2.841±0.01 |
| SNGP (Liu et al., 2020a)† | 79.9±0.03 | 49.0±0.02 | 0.025±0.012 | 0.117±0.014 | 0.847±0.01 | 2.626±0.01 |
| GSD (Tian et al., 2021) | 79.8±0.03 | 49.8±0.03 | 0.027±0.003 | 0.088±0.007 | 0.784±0.011 | **2.236**±**0.021** |
| Ours: $\alpha$-$\beta$ | 79.21±0.19 | 49.21±0.61 | **0.010**±**0.001** | **0.084**±**0.009** | 0.754±0.005 | 2.323±0.057 |

Table 3: **Calibration on CIFAR100** averaged over 5 seeds. † denotes results from Liu et al. (2020a).

As proved in Sec. 3.4, our model naturally leads to a calibration function in the family of *intra order-preserving functions* (Rahimi et al., 2020), which ensures good in-distribution calibration performance. Moreover, thanks to the improved sensitivity, the model can potentially achieve state-of-the-art out-of-distribution calibration as well. We benchmark our models on both clean CIFAR10/CIFAR100 as well as corrupted CIFAR10C/CIFAR100C (Hendrycks & Dietterich, 2019). Following prior works (Rahimi et al., 2020), we use 5-fold cross validation for in-distribution calibration experiments. As shown in Tab. 2 and Tab. 3, our model is comparable to state-of-the-art models on both in-distribution and out-of-distribution calibration. In the literature, OOD detection and calibration have been studied separately, but our method and experimentation tackles both tasks.

## 5 Conclusion

In this work, we propose to characterize the spectrum of out-of-distribution (OOD) data using covariate shift and concept shift. Unlike difficulty of OOD detection, distribution shift exposes intrinsic characteristics of OOD data. Consequently, to achieve good OOD detection performance, a model needs to consider both distribution shifts. At representation level, we derive two score functions that represent and capture each shift separately. At modeling level, inspired by these score functions, we propose a geometrically-inspired method, Geometric ODIN, to improve a model's sensitivity to both shift. Furthermore, Geometric ODIN is the first method that considers both OOD detection and calibration, targeting both concept and covariate shifts, and yields state-of-the-art performance in both tasks. Finally, we hope that the distribution shift perspective can lead to a new direction to study OOD data and unify the the fields of detection and calibration of OOD data.

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

# 6 APPENDIX

## 6.1 INTRODUCTION TO OUT-OF-DISTRIBUTION DETECTION

Out-of-Distribution detection is the task of separating in-distribution data from out-of-distribution (OOD) data. For example, a classifier, trained on a training distribution $P_{tr}(X, Y)$, is tested on a different test joint distribution $P_{tst}(X, Y)$. When the test joint distribution $P_{tst}(X, Y) \neq P_{tr}(X, Y)$, data sampled from $P_{tst}(X, Y)$ are considered OOD data. $P_{tst}(X, Y)$ can be different in two ways, i.e., covariate shift and concept shift (Sec. 3.2). Data sampled from the same distribution as training, $P_{tr}(X, Y)$, such as a conventional validation/test split, are in-distribution (ID) data. To detect OOD data, a score function is used such as maximum softmax probability (Hendrycks & Gimpel, 2016) and energy (Liu et al., 2020b). The score function assigns higher value to OOD data than ID data. Consequently, OOD data can be detected by setting a suitable threshold. The commonly used benchmark, Area Under the Receiver Operating Characteristic curve (AUROC), plots true positive rate of ID data against false positive rate of OOD data by sweeping through a range of thresholds, and is a threshold-independent metric for measuring OOD detection performance.

## 6.2 INTRODUCTION TO CONFIDENCE CALIBRATION

Colloquially, confidence calibration refers to aligning the maximum predicted probability (confidence) from a multi-class classifier to the empirical accuracy of the predicted class to be the ground truth. For example, if a classifier classifies 100 images to *dog* with a confidence of $0.8$ for each prediction, than 80 out of 100 those images should contain a dog. Formally, we adopt the definition from Kull et al. (2019) to define confidence calibration. Let $\hat{\mathbf{P}} : \mathcal{X} \to \Delta_x$ be a probabilistic multi-class classifier for $M$ classes. For any input $x \in \mathcal{X}$, $\hat{\mathbf{P}}(x) = (\hat{P}_1(x), ..., \hat{P}_M(x))$ is a probability vector.

**Definition 1** *A probabilistic classifier* $\hat{\mathbf{P}} : \mathcal{X} \to \Delta_x$ *is confidence-calibrated, if for any* $p \in [0, 1]$

$$P(Y = \arg \max \hat{\mathbf{P}}(x) | \max \hat{\mathbf{P}}(x) = p) = p$$

One notion of confidence calibration is the expected difference between the confidence and accuracy.

$$E_{\hat{P}} \left[ \left| P(Y = \arg \max \hat{\mathbf{P}}(x) | \max \hat{\mathbf{P}}(x) = p) - p \right| \right]$$

Expected Calibration Error (ECE) (Naeini et al., 2015) (Guo et al., 2017) is a metric that approximates this expectation. ECE partitions predictions into several $M$ equally-sized bins according to the predicted confidence, and then calculates average accuracy and confidence for each bin. Let $\hat{y}$ and $\hat{p}$ be the predicted class and confidence of an input $x$ with corresponding label $y$. $B_n$ denotes the $n$-th bin. The accuracy and confidence are calculated for each bin as follows:

$$acc(B_n) = \frac{1}{|B_n|} \sum_{i \in B_n} \mathbf{1}(\hat{y}_i = y_i) \tag{9}$$

$$conf(B_n) = \frac{1}{|B_n|} \sum_{i \in B_n} \hat{p}_i \tag{10}$$

Than ECE is defined as a weighted sum of the difference between confidence and accuracy in each bin. Therefore, lower ECE indicates better calibration.

$$ECE = \sum_{n=1}^{N} \frac{|B_n|}{N} |acc(B_n) - conf(B_n)|$$

## 6.3 GEOMETRIC SENSITIVITY DECOMPOSITION REVIEW

Geometric sensitivity decomposition (GSD) is proposed in Tian et al. (2021) to improve the sensitivity of neural networks to input changes. Specifically, GSD views the output of the last linear layer, i.e., logits, in a softmax-linear model as inner products, between the feature and weights in

the linear layer, , $\mathbf{l} \in \mathcal{R}^M =< \mathbf{f}, \mathbf{W} >$. The inner products can be written as products of norms and cosine similarity, e.g., the $i$th logit is $l_i^* = \|\mathbf{f}^*\|_2 \|\mathbf{w}_i\|_2 \cos \phi_i^*$. GSD proposes to decompose norms, $\|\mathbf{f}^*\|_2$, and angles, $\phi_i^*$, into two components: an instance-dependent residual component and an instance-independent scalar as shown in Eq. 11. The instance-independent scalar acts as freely optimizable parameters during training to minimize training loss.

$$
\begin{cases}
\|\mathbf{f}^*\|_2 = \|\mathbf{f}\|_2 + \mathcal{C}_f \\
|\phi_i^*| = |\phi_i| - |\mathcal{C}_\phi|
\end{cases}
\tag{11}
$$

Decomposed components in Eq. 11 can be plugged into equation for logits.

$$
\|\mathbf{f}^*\|_2 \cos \phi_i^* = \|\mathbf{f}^*\|_2 \cos |\phi_i^*| = (\|\mathbf{f}\|_2 + \mathcal{C}_f) \cos(|\phi_i| - |\mathcal{C}_\phi|)
\tag{12}
$$

$$
= (\|\mathbf{f}\|_2 + \mathcal{C}_f) \frac{1}{\cos |\mathcal{C}_\phi|} \cos |\phi_i| \left( 1 - \sin|\mathcal{C}_\phi|^2 \left( 1 - \underbrace{\frac{\cos |\mathcal{C}_\phi| \sin |\phi_i|}{\sin |\mathcal{C}_\phi| \cos |\phi_i|}}_{\text{E.q. } 13} \right) \right)
$$

Eq. 12 can be simplified by assuming that the angle $|\phi_i^*|$ is small. This equivalent assumes that training data are closely clustered to linear classifiers specified by the linear weight $\mathbf{W}$.

$$
\frac{\cos |\mathcal{C}_\phi| \sin |\phi_i|}{\sin |\mathcal{C}_\phi| \cos |\phi_i|} = \frac{\sin(|\phi_i| + |\mathcal{C}_\phi|) + \sin |\phi_i|}{\sin(|\phi_i| + |\mathcal{C}_\phi|) - \sin |\phi_y|} \approx 1
\tag{13}
$$

Therefore, logits can be written as the following.

$$
\|\mathbf{f}^*\|_2 \|\mathbf{w}_i\|_2 \cos \phi_i^* \approx \left( \underbrace{\frac{1}{\cos \mathcal{C}_\phi}}_{\alpha} \|\mathbf{f}\|_2 + \underbrace{\frac{1}{\cos \mathcal{C}_\phi}}_{\alpha} \overbrace{\mathcal{C}_f}^{\beta} \right) \|\mathbf{w}_i\|_2 \cos \phi_i
\tag{14}
$$

Because $\cos \mathcal{C}_\phi$ and $\mathcal{C}_f$ are instance-independent, we can parametrize them separately from the main network as $\alpha$ and $\beta$ respectively.

### 6.4 Intra Order Preserving Functions Review

Rahimi et al. (2020) formalizes a family of powerful calibration function: *intra order preserving functions*. Many familiar functions are in this family such as the softmax function and temperature scaling (Guo et al., 2017). Formally, a function $\mathbf{f} : \mathcal{R}^n \to \mathcal{R}^n$ is intra order-preserving if for any $x \in \mathcal{R}^n$, both $x$ and $\mathbf{f}(\mathbf{x})$ share share the same ranking. The following theorem from Rahimi et al. (2020) provides sufficient and necessary conditions for constructing an intra order preserving function.

**Theorem 1** *A continuous function* $\mathbf{f} : \mathcal{R}^n \to \mathcal{R}^n$ *is intra order-preserving, if and only if* $\mathbf{f}(\mathbf{x}) = S(x)^{-1} U \mathbf{w}(\mathbf{x})$ *with* $U$ *being an upper-triangular matrix of ones and* $\mathbf{w} : \mathcal{R}^n \to \mathcal{R}^n$ *being a continuous function such that*

- $\mathbf{w}_i(\mathbf{x}) = 0$ *if* $y_i = y_{i-1}$ *and* $i < n$,
- $\mathbf{w}_i(\mathbf{x}) > 0$ *if* $y_i > y_{i_1}$ *and* $i < n$,
- $\mathbf{w}_n(\mathbf{x})$ *is arbitrary*

*where* $y = S(x)x$ *is the sorted version (descending, i.e.,* $y_1 \geq y_2$*) of* $x$ *and* $S(x) : \mathcal{R}^n \to \mathcal{R}^n$ *is a sorting matrix.*

The task of constructing an intra order preserving function comes down to designing the function $\mathbf{w}(\mathbf{x})$. A specific form of $\mathbf{w}(\mathbf{x})$ is proposed in Rahimi et al. (2020): $\mathbf{w}_i(\mathbf{x}) = \sigma(y_i - y_{i-1})\mathbf{m}_i(\mathbf{x})$.

As long as $\sigma$ is a positive function, i.e., $\sigma(x) = 0$ when $x = 0$ and $\sigma(x) > 0$ otherwise, and $\mathbf{m}(\mathbf{x}) > 0$ is a *strictly* positive function, the contructed function will satisfy Theorem 1.

**Our construction of $\alpha$ and $\beta$ satisfies the necessary conditions.** For calibration, the function $\mathbf{f}$ acts on the logits $\mathbf{l}$ so we use $\mathbf{l}$ instead of $\mathbf{x}$ in subsequent proof. Let $l_i = \|\mathbf{f}\|_2 \|\mathbf{w_i}\|_2 \cos\phi_i$ denote the *sorted*[12] $i$th logit and $\bar{l}_i = \left(\frac{1}{\alpha(\mathbf{l})}\|\mathbf{f}\|_2 + \frac{\beta(\mathbf{l})}{\alpha(\mathbf{l})}\right)\|\mathbf{w_i}\|_2 \cos\phi_i$ detnote the actual output $i$th logit (*sorted*) with $\alpha$ and $\beta$. The goal is to show that $\bar{\mathbf{l}}$ and $\mathbf{l}$ share the same ranking by satisfying Theorem 1. We start by extracting the $\mathbf{m}(\mathbf{x})$ function in Eq. 15.

$$\bar{l}_i = \left(\frac{1}{\alpha(\mathbf{l})}\|\mathbf{f}\|_2 + \frac{\beta(\mathbf{l})}{\alpha(\mathbf{l})}\right)\|\mathbf{w_i}\|_2 \cos\phi_i = \underbrace{\left(\frac{1}{\alpha(\mathbf{l})} + \frac{\beta(\mathbf{l})}{\alpha(\mathbf{l})\|\mathbf{f}\|_2}\right)}_{m(\mathbf{l})}\overbrace{\|\mathbf{f}\|_2\|\mathbf{w_i}\|_2 \cos\phi_i}^{l_i} \quad (15)$$

Therefore, we can construct the $\mathbf{w}(\mathbf{x})$ function by taking the difference between two adjacent logits for $i < n$.

$$\begin{cases} \mathbf{w_i}(\mathbf{l}) = \bar{l}_i - \bar{l}_{i-1} = \underbrace{\left(\frac{1}{\alpha(\mathbf{l})} + \frac{\beta(\mathbf{l})}{\alpha(\mathbf{l})\|\mathbf{f}\|_2}\right)}_{m(\mathbf{l})>0}\overbrace{(l_i - l_{i-1})}^{\sigma \geq 0}, & \text{for} \quad i < n \\ \mathbf{w_i}(\mathbf{l}) = l_i & \text{for} \quad i = n \end{cases} \quad (16)$$

where $l_i \geq l_{i-1}$.

Because $0 < \alpha(\mathbf{l}) < 1$ and $\beta(\mathbf{l}) > 0$, the function $m(\mathbf{l})$ is strictly positive and thus satisfies Theorem 1 in Rahimi et al. (2020). Specifically, $\mathbf{m}(\mathbf{l})$ corresponds to the strictly positive function $\mathbf{m}(\mathbf{x})$ and $l_i - l_{i-1}$ corresponds to the positive function $\sigma(y_i - y_{i-1})$, due to the sorting in descending order.

## 6.5 Implementation Details

**Out-of-Distribution Detection** Following prior works (Liu et al., 2020a; Van Amersfoort et al., 2020; Mukhoti et al., 2021), we use Wide-ResNet-28-10 (Mukhoti et al., 2021) for all OOD detection experiments. We train the model using SGD and cross entropy loss with an initial learning rate of $0.1$ for 200 epochs. The SGD optimizer is configed with $5.0e - 4$ weight decay and $0.9$ momentum. Batch size is 128. The learning rate is annealed with a cosine scheduler (Loshchilov & Hutter, 2016). We adopt the official code[13] for the implementation of DDU (Mukhoti et al., 2021). DDU (Mukhoti et al., 2021) and Mahanobis (Lee et al., 2018) distance use density estimation by fitting a Gaussian mixture model (GMM) to the learned feature space. We use the official implementation of DDU to fit a GMM to the feature space (feature from the CNN). For Manhanobis distance, we fit a GMM to the low dimensional logit space instead for computational stability.

**Calibration** The backbone training follows the same procedure as the previous section. During the calibration/tuning stage, we train all models for 20 epochs with cosine learning rate decay using SGD and cross entropy loss. In this stage, we freeze the backbone models and only tune the calibration parameters/functions following (Guo et al., 2017). In our case, only $\alpha$ and $\beta$ are tuned. For calibration on out-of-distribution data, the models are tuned on the entire validation set. For calibration on in-distribution data, we use 5-fold cross validation using the validation set following prior work (Rahimi et al., 2020). For matrix scaling Guo et al. (2017) and Dirichlet scaling (Kull et al., 2019), we use the Off-Diagonal and Intercept Regularisation (ODIR) (Kull et al., 2019) with a default scaling of $1 \times 10^{-7}$. Note that we use the same initial learning rate, $0.1$, for all methods except for Dirichlet scaling. Dirichlet scaling requires smaller learning rate and we tuned the learning rate on different datasets. For CIFAR10, we use $0.01$ and for CIFAR100 we use $0.005$.

## 6.6 CIFAR100 Concept Shift Splits

While it is natural to associate covariate shift with increasing degrees of image corruption, finding a dataset to benchmark gradual concept shift is not straightforward because concept shift is traditionally thought as binary: overlapping or non-overlapping. However, not all non-overlapping labels

---

[12]The original order can be restored by applying $S(\mathbf{l})^{-1}$ to $\mathbf{l}$.

[13]https://github.com/omegafragger/DDU

|  | 1 | 2 | 3 | 4 | 5 | 6 | 7 | 8 | 9 | 10 | AVE. | STD. |
|---|---|---|---|---|---|---|---|---|---|---|---|---|
| CIFAR10 | airplane | automobile | bird | cat | deer | dog | frog | horse | ship | truck | | |
| Group 9 | cattle | shrew | motorcycle | squirrel | snake | trout | sea | tractor | bus | pickup | 24.96 | 2.39 |
| Group 8 | bear | elephant | leopard | camel | lizard | rabbit | beaver | spider | raccoon | orchid | 21.99 | 0.44 |
| Group 7 | lion | mountain | crab | bicycle | turtle | beetle | train | mouse | snail | otter | 20.18 | 1.14 |
| Group 6 | possum | shark | forest | pine | dinosaur | boy | porcupine | wolf | road | butterfly | 17.79 | 0.32 |
| Group 5 | girl | rocket | man | tiger | bee | tank | whale | baby | kangaroo | dolphin | 16.26 | 0.44 |
| Group 4 | willow | worm | chimpanzee | skunk | cup | mushroom | oak | cockroach | crocodile | hamster | 14.64 | 0.55 |
| Group 3 | castle | can | bridge | lobster | house | bed | fox | maple | pear | woman | 12.65 | 0.63 |
| Group 2 | palm | streetcar | pepper | keyboard | bottle | seal | rose | couch | caterpillar | goldfish | 10.18 | 0.51 |
| Group 1 | flatfish | apple | orange | plate | table | tulip | bowl | television | skyscraper | ray | 8.95 | 0.25 |
| Group 0 | wardrobe | lamp | plain | lawnmower | chair | poppy | clock | cloud | sunflower | telephone | 7.5 | 0.97 |

Table 4: **CIFAR100 Concept Shift Splits** Small group numbers indicate less conceptual similarity to CIFAR10 classes. The similarity is calculated using inner product between the Glove embeddings of a CIFAR100 class and a CIFAR10 class. For each CIFAR100 class, the largest similarity to each CIFAR10 class is taken as the overall similarity to CIFAR10. The average shows average similarity to CIFAR10 and the standard deviation shows in-group variance.

|  |  | AUROC↑ | | | | | TNR@TPR95↑ | | | | |
|---|---|---|---|---|---|---|---|---|---|---|---|
|  |  | 1 | 2 | 3 | 4 | 5 | 1 | 2 | 3 | 4 | 5 |
| $g(x)$ | vanilla | 58.66±0.93 | 64.59±1.42 | 69.17±1.90 | 69.18±2.03 | 72.66±2.44 | 10.75±0.50 | 17.31±0.50 | 24.05±1.12 | 24.13±1.24 | 29.88±2.03 |
|  | $\alpha(x)$-only | 62.01±1.03 | 69.60±1.55 | 75.29±1.90 | 75.46±1.91 | 79.50±2.32 | 9.21±1.86 | 14.04±3.97 | 19.39±5.64 | 19.56±5.66 | 24.38±6.88 |
|  | $\alpha(x)$-$\beta(x)$ | **67.00±0.55** | **77.87±0.51** | **85.77±0.53** | **85.86±0.53** | **91.25±0.56** | **14.07±1.50** | **26.32±2.55** | **41.77±3.03** | **42.33±3.11** | **57.99±2.83** |
| $h(y,x)$ | vanilla | 61.02±0.44 | 69.80±0.74 | 76.43±0.79 | 76.47±0.81 | 81.39±0.90 | 9.98±0.70 | 16.84±1.15 | 24.70±1.53 | 24.61±1.78 | 32.13±2.08 |
|  | $\alpha(x)$-only | **63.98±0.88** | **74.39±0.97** | **81.58±0.91** | **81.66±0.93** | **86.67±0.85** | **10.52±1.07** | **18.18±2.20** | **27.69±3.68** | **27.65±3.96** | **36.79±5.43** |
|  | $\alpha(x)$-$\beta(x)$ | 59.11±0.27 | 68.43±0.78 | 76.19±0.85 | 76.16±0.85 | 82.20±0.73 | 10.22±0.23 | 17.72±0.85 | 27.28±1.32 | 27.19±1.26 | 36.91±1.55 |
| $\mathcal{U}$ | vanilla | 60.66±0.51 | 68.42±0.65 | 74.32±0.90 | 74.34±1.00 | 78.72±1.21 | 10.36±0.37 | 17.60±0.67 | 26.08±0.72 | 25.97±1.10 | 33.69±1.24 |
|  | $\alpha(x)$-only | 64.20±0.86 | 74.05±1.02 | 81.16±0.98 | 81.29±0.99 | 86.18±0.97 | 10.25±0.50 | 17.63±1.08 | 26.65±1.93 | 26.62±2.15 | 35.57±3.52 |
|  | $\alpha(x)$-$\beta(x)$ | **66.86±0.47** | **77.92±0.45** | **85.82±0.52** | **85.83±0.47** | **91.10±0.47** | **13.32±1.13** | **25.21±2.12** | **40.67±3.13** | **40.74±2.65** | **56.48±2.62** |

Table 5: **Capturing Covariate Shift (Motion Blur)** All results are averaged over 5 runs.

are the same. For example, *pickup truck* (CIFAR100) is much closer to *truck* (CIFAR10) than *sunflowers* (CIFAR100) is semantically. To create this gradual concept/semantic shift, we propose to divide the CIFAR100 dataset into 10 sub-datasets with increasing conceptual difference from CIFAR10 classes. Specially, we use Glove word embeddings (Pennington et al., 2014) wtih 6 billion tokens trained on the entire wikipedia2014 and Gigaword5 (Napoles et al., 2012) to measure semantic closeness (inner product) between CIFAR100 and CIFAR10 classes. The result is 10 subdatasets split from CIFAR100 as shown in Tab. 4. Our experiments demonstrate that the conceptual difference measured by semantic similarity in language translates to a spectrum of near OOD datasets measured by the difficulty of OOD detection in the image space.

## 6.7 ADDITIONAL RESULTS

We provide additional results and tabulated results of figures in the main paper. Specifically, Tab. 5 and Tab. 8 are tabulated versions of Fig .3 and Fig. 4 in the main paper respectively. In addition to motion blur, we also provide out-of-distribution detection results on shot noise in Tab. 7 and zoom blur in Tab. 6.

## 6.8 EXTENDED RELATED WORK

**Out-of-Distribution (OOD) detection** methods can be largely divided into two camps depending on whether they require OOD data during training. Hendrycks et al. (2018) leverages anomalous

|  |  | AUROC↑ | | | | | TNR@TPR95↑ | | | | |
|---|---|---|---|---|---|---|---|---|---|---|---|
|  |  | 1 | 2 | 3 | 4 | 5 | 1 | 2 | 3 | 4 | 5 |
| $g(x)$ | vanilla | 61.28±1.13 | 67.27±1.71 | 72.82±2.27 | 77.22±2.69 | 81.84±3.16 | 13.45±0.40 | 19.58±1.02 | 27.45±2.32 | 34.91±4.21 | 44.26±6.32 |
|  | $\alpha(x)$-only | 66.64±1.59 | 68.72±2.82 | 72.16±3.87 | 73.84±5.01 | 75.84±6.29 | 10.18±1.70 | 10.70±3.02 | 13.38±4.92 | 14.84±6.35 | 17.02±8.37 |
|  | $\alpha(x)$-$\beta(x)$ | **71.21±5.89** | **77.72±2.77** | **81.78±6.68** | **85.53±7.40** | **89.74±7.36** | **20.96±5.50** | **26.17±8.08** | **35.67±11.38** | **43.35±14.53** | **54.86±18.97** |
| $h(y,x)$ | vanilla | 66.19±0.78 | 72.12±1.44 | 78.67±1.67 | 83.14±1.81 | 87.83±1.65 | 13.43±0.88 | 18.11±1.57 | 26.31±2.84 | 34.37±4.00 | 45.88±4.64 |
|  | $\alpha(x)$-only | **69.34±1.63** | **75.49±2.40** | **81.93±2.61** | **86.37±2.55** | **90.65±2.28** | **15.69±1.34** | **22.16±2.80** | **32.81±4.52** | **42.91±6.14** | **55.99±7.39** |
|  | $\alpha(x)$-$\beta(x)$ | 63.73±1.26 | 69.92±2.29 | 74.80±2.31 | 79.42±3.61 | 85.11±4.50 | 13.73±1.07 | 18.40±1.67 | 27.09±2.53 | 35.60±2.81 | 48.24±2.39 |
| $\mathcal{U}$ | vanilla | 64.63±0.50 | 70.94±0.88 | 77.26±1.12 | 81.87±1.45 | 86.64±1.54 | 13.95±0.50 | 19.35±1.18 | 28.27±2.01 | 36.98±3.00 | 48.82±3.69 |
|  | $\alpha(x)$-only | 69.65±1.11 | 74.51±1.67 | 80.42±1.88 | 84.46±2.00 | 88.68±1.86 | 14.65±0.80 | 19.19±1.50 | 27.64±2.55 | 35.07±3.81 | 45.66±5.12 |
|  | $\alpha(x)$-$\beta(x)$ | **71.25±5.14** | **78.00±1.74** | **82.23±5.71** | **86.14±6.44** | **90.50±6.41** | **19.93±3.05** | **25.84±4.08** | **36.81±5.72** | **46.54±7.28** | **60.70±8.87** |

Table 6: **Capturing Covariate Shift (Zoom Blur)** All results are averaged over 5 runs.

| | | AUROC↑ | | | | | TNR@TPR95↑ | | | | |
|---|---|---|---|---|---|---|---|---|---|---|---|
| | | 1 | 2 | 3 | 4 | 5 | 1 | 2 | 3 | 4 | 5 |
| $g(x)$ | vanilla | 59.84±1.81 | 65.04±2.43 | 73.34±3.47 | 75.85±3.97 | 79.53±4.80 | **13.08±2.86** | **18.59±5.03** | 28.40±11.06 | 32.46±14.24 | 38.75±20.04 |
| | $\alpha(x)$-only | 61.88±2.75 | 67.05±3.65 | 75.05±4.64 | 77.35±4.66 | 80.49±4.49 | 8.48±1.75 | 10.27±2.85 | 12.98±5.77 | 13.82±7.16 | 15.42±9.38 |
| | $\alpha(x)$-$\beta(x)$ | **64.78±0.99** | **73.21±1.84** | **87.01±3.05** | **90.63±2.99** | **94.55±2.42** | 11.34±1.60 | 18.58±4.55 | **44.31±12.91** | **55.61±14.72** | **71.07±14.96** |
| $h(y,x)$ | vanilla | 68.14±1.72 | 75.80±2.32 | 85.92±3.13 | 88.20±3.24 | 90.97±3.44 | 16.93±1.80 | 24.75±3.08 | 40.55±6.21 | 45.58±7.59 | 52.06±10.06 |
| | $\alpha(x)$-only | **69.88±0.79** | **78.30±0.93** | **89.39±1.04** | **91.79±1.14** | **94.49±1.23** | **18.64±1.12** | **28.91±1.36** | **52.12±3.39** | **59.56±4.80** | **69.62±7.44** |
| | $\alpha(x)$-$\beta(x)$ | 68.62±0.86 | 76.47±1.00 | 87.26±1.01 | 90.00±1.40 | 93.42±1.96 | 17.41±0.52 | 26.48±1.73 | 46.80±4.93 | 54.26±7.29 | 65.65±11.14 |
| $\mathcal{U}$ | vanilla | 65.07±2.17 | 72.59±2.29 | 83.05±2.27 | 85.62±2.34 | 88.95±2.84 | **16.32±1.11** | 23.96±1.95 | 39.06±3.77 | 44.28±5.13 | 51.05±7.28 |
| | $\alpha(x)$-only | **67.81±1.22** | 75.91±1.38 | 87.19±1.61 | 89.81±1.71 | 92.96±1.74 | 15.07±0.82 | 22.39±1.84 | 40.17±5.87 | 47.07±7.81 | 57.90±11.05 |
| | $\alpha(x)$-$\beta(x)$ | 67.23±0.55 | **76.13±0.98** | **89.12±2.02** | **92.19±2.12** | **95.49±1.89** | 14.94±0.97 | **24.27±3.25** | **51.27±9.38** | **61.50±11.07** | **75.45±11.63** |

Table 7: **Capturing Covariate Shift (Shot Noise)** All results are averaged over 5 runs.

| | | AUROC↑ | | | | | | | | | | TNR@TPR95↑ | | | | | | | | | |
|---|---|---|---|---|---|---|---|---|---|---|---|---|---|---|---|---|---|---|---|---|---|
| | | 1 | 2 | 3 | 4 | 5 | 6 | 7 | 8 | 9 | 10 | 1 | 2 | 3 | 4 | 5 | 6 | 7 | 8 | 9 | 10 |
| $g(x)$ | vanilla | 75.88 | 79.64 | 80.35 | 81.43 | 80.23 | 81.48 | 79.83 | 80.03 | 78.99 | 83.37 | **33.12** | 38.52 | 41.94 | 41.38 | **40.42** | 44.20 | 39.34 | 42.18 | 38.72 | 47.06 |
| | $\alpha(x)$ | **80.28** | 87.23 | 86.77 | 89.10 | 87.37 | 89.59 | 86.90 | 86.27 | 85.66 | 87.87 | 29.34 | 38.42 | 37.14 | 44.28 | 36.90 | 44.40 | 37.32 | 38.80 | 32.22 | 43.00 |
| | $\alpha(x)$-$\beta(x)$ | 79.05 | **89.50** | **88.51** | **90.49** | **88.97** | **90.98** | **89.29** | **90.98** | **91.78** | **91.15** | 31.20 | **42.18** | **41.02** | **49.14** | 40.16 | **50.76** | **42.94** | **52.00** | **49.16** | **51.32** |
| $h(y,x)$ | vanilla | 86.78 | 90.74 | 91.00 | 90.20 | 89.85 | 91.10 | 91.84 | 91.85 | 93.90 | 94.09 | 46.16 | 52.50 | 52.58 | 51.66 | 48.58 | 52.64 | 53.38 | 58.10 | 61.60 | 63.28 |
| | $\alpha(x)$ | 87.83 | **91.36** | **91.68** | **91.36** | **91.43** | **92.43** | **92.75** | **93.58** | **94.87** | **95.52** | 47.46 | **53.78** | 55.46 | **55.90** | 53.44 | **61.04** | **58.38** | **65.60** | **68.92** | **72.54** |
| | $\alpha(x)$-$\beta(x)$ | **88.16** | 89.93 | 90.18 | 89.68 | 89.53 | 91.31 | 91.81 | 92.59 | 94.41 | 95.22 | **48.74** | 51.66 | 53.74 | 53.88 | 52.86 | 59.32 | 57.70 | 64.56 | 68.10 | 72.54 |
| $\mathcal{U}$ | vanilla | 84.02 | 88.52 | 88.45 | 88.59 | 87.52 | 89.09 | 89.18 | 88.96 | 91.13 | 92.26 | **46.78** | 52.96 | 54.38 | 53.38 | 50.28 | 54.02 | 53.88 | 58.04 | 60.02 | 63.40 |
| | $\alpha(x)$ | **86.54** | 91.84 | **91.99** | 92.56 | **92.06** | 93.26 | **92.69** | 93.13 | 94.12 | **95.10** | 46.68 | 54.48 | **56.30** | 57.82 | **53.92** | 62.20 | **57.88** | 62.86 | 64.56 | **70.08** |
| | $\alpha(x)$-$\beta(x)$ | 84.58 | **91.96** | 91.85 | **92.65** | 91.94 | **93.31** | 92.55 | **93.54** | **95.06** | 95.07 | 44.72 | **54.62** | 55.10 | **58.94** | 52.76 | **63.54** | 57.72 | **66.42** | **69.20** | 70.02 |

Table 8: **Capturing Concept Shift (CIFAR100 Splits)** All results are averaged over 5 runs.

data in training which enables classifiers to generalize and detect unseen OOD data. Thulasidasan et al. (2020) extends existing classifier to include an OOD class. Roy et al. (2021) assigns multiple classes for outliers instead of a single class. Our method belongs to the class of methods that do not assume the availability of OOD data during training. Hendrycks & Gimpel (2016) discovers that correctly classified examples have larger maximum softmax probability (MSP) and propose to use it to detect incorrect predictions and OOD data. Lee et al. (2018) proposes to use Mahalanobis distance by fitting a Gaussian mixture model (GMM) in the feature space. Mukhoti et al. (2021) uses log density of the GMM model instead. Liu et al. (2020b) uses an energy score as the uncertainty metric to distinguish between in-distribution and OOD data. ODIN (Liang et al., 2017) uses a combination of input processing and post-training tuning to improve OOD detection performance. Generalized ODIN Hsu et al. (2020) (also Techapanurak et al. (2019)) includes an additional network in the last layer to improve OOD detection during training. There are many other interesting OOD detection approaches that have achieved state-of-the-art performance without OOD data such as using contrastive learning with various transformations (Winkens et al., 2020; Tack et al., 2020), training a deep ensemble of multiple models (Lakshminarayanan et al., 2016) and leveraging large pretrained models (Fort et al., 2021). They require extended training time, hyperparameter tuning and careful selections of transformations, whereas our method does not introduce any hyperparameters and has negligible influence on standard cross-entropy training time. For example, Winkens et al. (2020) trains a constrastive model for 1200 epochs on CIFAR100 whereas our model requires only 200 epochs (same as standard classifier training) on the same dataset.

**Model Calibration** methods can also be largely divided in two categories: 1) training time calibration using augmentations (Thulasidasan et al., 2019; Jang et al., 2021), using modified losses (Kumar et al., 2018); 2) post-hoc calibration (Guo et al., 2017; Kull et al., 2019; Kumar et al., 2019; Rahimi et al., 2020). Guo et al. (2017) proposes to calibrate the confidence of a trained classifier using temperature scaling, a single scalar that softens overconfident softmax predictions. Kull et al. (2019) extends single-class confidence calibration to multiclass calibration using the Dirichlet distribution. Kumar et al. (2019) proposes a scaling-binning calibrator that is more sample efficient. Recently, Rahimi et al. (2020) formally generalizes a family of expressive functions for calibration, *the intra order-preserving functions*. This class of functions has more representation power to calibrate more complex decision boundaries in neural networks. Our proposed method belongs to this class of functions. However, all these works focus on calibration of in-distribution data. To obtain better calibration on OOD data, on which the confidence of a model needs to decrease accordingly, sensitivity and deterministic uncertainty modeling is explored by SNGP (Van Amersfoort et al., 2020) and DUQ (Liu et al., 2020a). SNGP uses a bounded spectral normalization regularization (Miyato et al., 2018) during training and DUQ adopts a two-sided gradient penalty (Gulrajani et al., 2017) to improve model sensitivity to distribution shift. Our proposed model not only belongs to the family of intra order-preserving functions, which ensures good in-distribution calibration, but also improves sensitivity to distribution shift, which improves out-of-distribution calibration simultaneously.

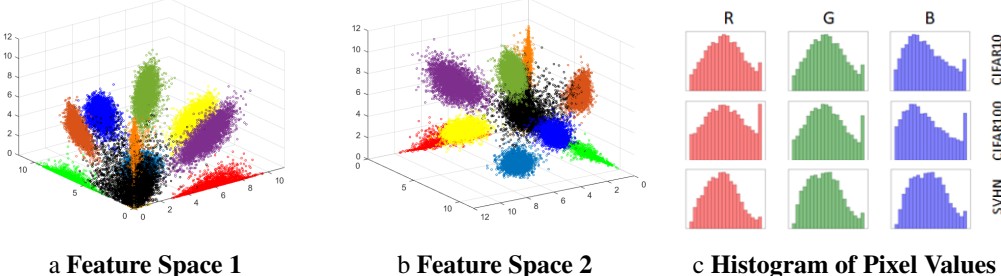

a **Feature Space 1**    b **Feature Space 2**    c **Histogram of Pixel Values**

Figure 5: Fig. 5a and Fig. 5b show visualization of a 3-dimensional feature space of a model trained on CIFAR10. The black cluster represents Gaussian-noise corrupted CIFAR10 data. Fig. 5c shows histogram of pixels of CIFAR10, CIFAR100 and SVHN.

## 7    DISCUSSION ON COVARIATE AND CONCEPT SCORE DECOMPOSITION

In Sec. 3.2 in the main paper, two score functions are proposed to capture changes due to covariate and concept shift. As shown in Eq. 17, $g(x)$ denotes the **covariate shift score function** and $h(y, x)$ denotes the **concept shift score function**.

$$\mathcal{U} = \max_j l_j - \frac{1}{M} \sum_{i=1}^{M} l_i = \overbrace{\|\mathbf{f}\|_2}^{g(x)} \underbrace{\left( \max_j \|\mathbf{w}_j\|_2 \cos \phi_j - \frac{1}{M} \sum_{i=1}^{M} \|\mathbf{w}_i\|_2 \cos \phi_i \right)}_{h(y,x)} \quad (17)$$

Conceptually, these two scores disentangle covariate shift and concept shift. However, in practice, both shifts almost always happen at the same time in the image domain, i.e., covariate shifted data, e.g., noised data, often result in concept shift, i.e., increasing ambiguity in class assignment. Therefore, both scores can increase and decrease simultaneously as observed in Sec. 4.1.2 and Sec. 4.1.3. The importance of separating them conceptually is to provide a clean perspective to study robust out-of-distribution detection methods that will work well on different situations under either a single distribution shift or a mixed of shifts. It very likely that better score functions can be derived to better disentangle the effects of distribution shifts. In turn, methods that improve sensitivity of those new score functions can be motivated.

## 8    ADDITIONAL FIGURES

While it seems impossible that concept shift could happen without change in $P(x)$, this is possible if we define the support, $P(x)$, as the distribution of pixels, ie., the lowest level characteristics, completely stripped of any semantic information. A simple example would be reshuffling the pixels of a CIFAR10 image where the semantics, $P(y|x)$, is completely changed, while $P(x)$ remains the same. Using CIFAR100 to represent concept shift is based on this reasoning. We provide a visualization of pixel distributions of CIFAR10, CIFAR100 and SVHN in Fig. 5c. CIFAR10 and CIFAR100 share very similar pixel distributions while the pixel distributions of SVHN are very different from those of CIFAR10. The similarity in pixel distributions between CIFAR10 and CIFAR100 could be attributed to the data generation/collection process.

To further support the statement that norms are sensitive to covariate shift and justify the choice of defining the covariate shift score $g(x) = \|x\|_2$, we show a visualization of CIFAR10 classes in the feature space of a model trained on CIFAR10 in Fig. 5a and 5b. Similar to Liu et al. (2018), we change the last feature dimension of a ResNet to 3. In this way, we can directly visualize the learned feature space without resorting to dimension reduction techniques such as T-SNE (Wattenberg et al., 2016). In addition to the clean validation data from CIFAR10, we also visualize CIFAR10 data corrupted by Gaussian noise. The noised data represents severe covariate shift. We can observe that the noised data are clustered around the origin indicating very small norms.

