# OpenReview forum: "Exploring Covariate and Concept Shift for Detection and Confidence Calibration of Out-of-Distribution Data"
_ICLR.cc/2022/Conference — ICLR 2022 Submitted_

### Official Review · Reviewer_kVK3 · 2021-10-27

**Correctness:** 3
**Technical Novelty And Significance:** 3
**Empirical Novelty And Significance:** 3
**Recommendation:** 5
**Confidence:** 4

**Main Review:**

**Strengths**

- The paper provides a novel perspective about the KL divergence and the softmax linear classifier, and how they can be turned into a sensible OOD detection score function that works well. I like the proposed formulation and the interpretation presented in Sec.3.2, which gives the community some new insights.

- The fields of OOD detection and confidence calibration are often studied separately though they are closely related to each other. The paper offers a unified approach for them, which is a good contribution to the literature.

- The new CIFAR 100-based dataset is also a good contribution, which would be of interest to the community.

**Weaknesses**

The major weakness is in the experiments on the covariate shift datasets (i.e., CIFAR 10C/100C) and how the findings are presented. Covariate shift is something we want the model to overcome in practice, which has been extensively studied in the OOD generalization (or domain generalization) literature. The paper merely focuses on distinguishing between ID and OOD data resulting from covariate shift, which might send a misleading message that contradicts with the field of OOD generalization. CIFAR 10C and 100C have been widely used to test model robustness. The paper should also report the accuracy numbers to justify whether the proposed approach (and the compared ones) still maintain the discriminative power. Such a study shouldn't be isolated from OOD detection.

**Questions**

Some places are unclear and need further clarifications.

1. Is it necessary to use a linear layer to learn the alpha/beta in GSD? What is the motivation for such a modification? And what if the original GSD formulation is used?

2. In Sec.3.4, I don’t fully understand why having a positive function helps calibration. Appx.6.4 proves that the proposed formulation preserves the order of logits when updating the parameters but does not explain why doing so helps calibrate the confidence values. Also, would it be possible to provide the confidence-vs-accuracy figure and the metrics as used in [a]?

3. What are alpha-beta, alpha-only and beta-only? There is no explanation for these variants. I’d like to confirm if alpha-only means only the alpha in Eq.7 is used while the beta is set to a constant?

4. In Sec.4.1.2, the statement of “covariate shift does not happen in isolation from concept shift” seems inaccurate. Covaraite shift can certainly occur without shift in the label space, like changes in lighting or viewpoint that create a divergence in the input marginal distribution but do not change the labels anyway. Maybe I misunderstood the sentence?

[a] Guo, C., Pleiss, G., Sun, Y., & Weinberger, K. Q. (2017, July). On calibration of modern neural networks. In International Conference on Machine Learning (pp. 1321-1330). PMLR.

**Summary Of The Paper:**

The paper starts from the KL divergence between a uniform distribution and the predicted distribution, based on which two score functions are derived for covariate shift and concept shift, respectively. The covariate shift score measures the feature norm while the concept shift score is essentially the difference between the cosine distance of the predicted class and the average cosine distance of all classes.

Furthermore, the paper integrates a variant of the geometric sensitivity decomposition method introduced in a recent work into the proposed score functions, aiming to make them more sensitive to distribution shift and to help calibrate the model as well.

A new dataset called CIFAR 100 Splits is also built, which contains 10 sub-datasets with an increasing level of concept shift compared to CIFAR 10.

The results show that the proposed approach achieves decent performance in OOD detection (in both the scenarios of covariate shift and concept shift) as well as calibration.

**Summary Of The Review:**

Overall, I like the technical contributions, which have been proved effective and provide new insights to the community. But I am concerned with the weakness part mentioned above, which should be addressed. I give borderline accept for now and will reassess the paper after the discussion period.

---

> ### Comment · Reviewer_kVK3 · 2021-11-26
> **Post-rebuttal comments**
>
> Thanks for the detailed response.
>
> My major concern about the use of CIFAR10C/100C is unresolved. These two datasets contain the same classes as the training dataset so treating them as OOD is inconsistent with the widely adopted "norm" in the OOD detection literature [a, b], i.e., to detect abnormal data not belonging to training classes. *In the context of OOD detection*, it makes more sense to define "near-OOD" as those having similar background/style with training images but belonging to novel classes.
>
> Though I like the technical contributions, the paper isn't ready for publication at its current form. I'd suggest the authors re-define the term for "near-OOD" and remove CIFAR10C/100C from the experiments.
>
> [a] Semantically Coherent Out-of-Distribution Detection. ICCV'21.
>
> [b] MOOD: Multi-level Out-of-distribution Detection. CVPR'21.

---

### Official Review · Reviewer_rWjW · 2021-11-01

**Correctness:** 3
**Technical Novelty And Significance:** 3
**Empirical Novelty And Significance:** 2
**Recommendation:** 5
**Confidence:** 5

**Main Review:**

Positives:

- The paper is very well written, with a clear presentation and writing flow. I enjoyed reading through it.

- To simulate the gradual concept shift, the paper proposes a new benchmark CIFAR100 Splits using word-embeddings.

- The papers compared with a broad spectrum of baseline methods on OOD detection and calibration.

Concerns and questions for the authors:

- First and foremost, while the need for detecting semantic shifts is clear, the motivation of detecting covariate shifts confusing and questionable w.r.t the existing literature. In the former case, the goal is to **abstain from making a prediction** on something that the model's class space does not support, which is natural in real-world applications. However, in the case of covariate shift (say CIFAR10C), wouldn't it be desirable to generalize to the shifted data? This seems to be commonly formulated and tackled as a robustness and OOD generalization problem rather than an OOD detection problem. If the model is indeed robust and generalizable to the covariate shift, one would expect the OOD score statistics to be almost indistinguishable between the clean and corrupted data. However, the paper seems to challenge otherwise. The contribution and evaluation would have a much clearer and well-defined focus with near OOD and far OOD (on the semantic side).

- It'd be great if the authors can provide further justification on why the feature norm (the g(x) after decomposition) captures the covariate shift but not semantic shift? As the core contribution, this score decomposition (for representing covariate shift vs semantic shift) isn't rigorously justified. It'd be interesting to also measure the feature norm and see the separability between ID and semantic OOD. Table 1 can be expanded to include g(x) and U for the vanilla model (trained with cross-entropy), alpha model as well. Based on Table 1, it seems like g(x) is also effective in distinguishing semantic OOD (SVHN), which weakens the argument and covariate-semantic score decomposition.

- A missing baseline [1] also utilizes the cosine similarity for OOD detection, which shows strong performance together with generalized ODIN [2]. The paper can strengthen its SOTA claim by performing a comparison with these two methods.

- The authors could have provided an additional evaluation on other datasets like Texture, LSUN, and Places 365 similar to previous works in OOD detection literature (as listed in footnote 1). Together with CIFAR10-CIFAR100 evaluation, this would make the experiments more comprehensive.

- The definition of U in Equation (4) seems to coincide with the ODIN score, which also showed that the separability between ID and OOD can be approximated as $\frac{1}{M}\sum_i^M [\max l_j - l_i]$ (under a large temperature). This connection might be interesting to draw explicitly in the revision.

- The authors should clearly define the concept of "out-of-distribution calibration". The notion of calibration commonly refers to making the posterior matching the empirical accuracy on the in-distribution data. Why is it necessary to calibrate for OOD data, if the model shouldn't make an erroneous prediction anyway? For example, given the CIFAR-10 trained model, what does calibration error mean for an SVHN input?

- Code is missing so the reproducibility is unclear.

Minor:
- Page 3, "Concept shift appears when P_tr(Y |X)= P_tst(Y |X) and P_tr(X) = P_tst(X)"---> this is not correct as concept shift would imply the input shifted accordingly.

[1] E. Techapanurak, M. Suganuma, T. Okatani. Hyperparameter-Free Out-of-Distribution Detection Using Cosine Similarity.

[2] Y. Hsu, Y. Shen, H. Jin, Z. Kira. Generalized ODIN: Detecting Out-of-Distribution Image Without Learning From Out-of-Distribution Data. CVPR 2020.






**Summary Of The Paper:**

This paper proposes a method for detecting two types of distributional shifts: covariate shifts in the input space $\mathcal{X}$ (due to input corruption) and semantic shifts (due to test data falling outside the support set of ID classes, $ y_\text{test} \notin \mathcal{Y}$). The idea is based on the decomposition of KL-divergence between softmax prediction and a uniform vector. Furthermore, the authors propose Geometric ODIN to improve OOD detection and calibration, outperforming strong baseline on CIFAR10, CIFAR 100, and SVHN datasets.

**Summary Of The Review:**

Overall the paper is well written, with comprehensive evaluations. However, the reviewer is concerned about several conceptual confusion that the paper may introduce (as raised in my detailed comments above). The reviewer hopes to see the satisfactory changes made, and increase the score accordingly.

---

### Official Review · Reviewer_kPST · 2021-11-02

**Correctness:** 2
**Technical Novelty And Significance:** 2
**Empirical Novelty And Significance:** 2
**Recommendation:** 3
**Confidence:** 4

**Main Review:**

The overall aim of the submission to explore OOD detection for both covariate shift as well as concept shift (along a spectrum of increasing differences in concepts) is interesting and thought-provoking. However, I’m slightly confused by how concept shift is defined: $P_{tr}(Y | X) \neq P_{tst}(Y | X)$ but $P_{tr}(X) = P_{tst}(X)$? How can the distributions over X be the same if different concepts are being represented in tr and tst? This would only make sense if the exact same objects are being redefined significantly over the same support in different domains, which is not quite what is being explored in this paper, as far as I can tell (different objects are showing up in the test OOD set). In the context of this paper, I’d recommend simply describing concept shift as cases when the concepts have shifted but the more superficial characteristics are unchanged, such as low-level image  statistics (as discussed earlier in [1,2]). (This issue does not apply for covariate shift, since the question there is would you apply the same label to images with the same object but different styles, so we can have the same $P(Y|X)$ with shift in $P(X)$.)


While the score functions are described as being derived “theoretically”, it seems that the use of the norm-score does not really have any theoretical backing (that I could spot), instead it is claimed that “Intuitively, activation of a neural network on covariate-shifted data should be weaker”. I’m not sure I fully understand this intuition; it seems to me that one cannot say what would happen to the activations of a neural network on covariate-shifted data; in fact, I imagine one could easily synthesize some sort of adversarial example to maximally activate neurons with data that is highly covariate-shifted from the in-distribution. Also, if the derivation of U is based so strongly upon sandwiching the KL, why not simply use the KL as a score?


I don’t fully understand the reasoning behind making alpha and beta functions of f. I followed the initial motivation that a decoupling of norms and angles might help, and that an existing method has shown how to perform that decoupling. But there isn’t a clear explanation for why alpha, beta needed to be functions of f, and I couldn’t think of one either; it seemed more reasonable to expect, based on the motivation presented in the submission, that these should be instance-independent. Could you unpack the reason in the rebuttal?


I’d recommend rewording some of the framings of novelty such as “existing works do not distinguish between different types of OOD datasets”: Hsu et al. [1] discussed the distinction between “non-semantic” and “semantic” shift, which as far as I can tell, is pretty much synonymous with the paper’s “covariate” and “concept” shift terminology, but without the notational difficulties discussed above. Ahmed et al. [2] discussed the need for detecting “semantic” anomalies, since we typically want to be robust to non-semantic shift rather than flag it down (which is also part of the reason benchmarks such as corrupted-CIFAR are typically used to test for robustness rather than as OOD detection). The novel evaluation perspective in the submission, which is quite interesting, is the breakdown of CIFAR-100 into sets of classes with increasing semantic shift.


In the experiments, it is not clear why quite a few of the numbers deviate significantly from those reported elsewhere. For example, [3] reported the Mahalanobis method to score 98.4 AUROC for CIFAR100/SVHN, while the submission reports 84.4. The CIFAR10/SVHN performance is reported in [3] as 99.1 for Mahalanobis, while the submission reports 93.1. The proposed method is characterized as achieving “state-of-the-art results”, but it appears from works such as [4] that the Mahalanobis baseline and the Gram method often outperform the proposed method (the original papers developing the method also report similar numbers). If the difference is due to differences in architectural/training choices, these choices need to be motivated since most of the reported baseline numbers in the submission seem significantly worse than in existing literature.

[1] Generalized ODIN, Hsu et al.

[2] Detecting semantic anomalies, Ahmed et al.

[3] Detecting OOD examples with gram matrices, Sastry et al.


**Summary Of The Paper:**

The submission proposes to evaluate covariate shift and "concept shift" through two scoring functions, one based on feature norms (for covariate shift) and the other based on angles (for concepts). To help the network decouple norms and angles, (a modified version of) a very recent method is used to decompose the feature norm and the cosine-distances and also calibrate the network. Experimental results appear to demonstrate good performance on both near and far OOD examples, as well as on calibration metrics.


**Summary Of The Review:**

While the submission is certainly intriguing, promoting further the decoupling of semantic and non-semantic distributional shift detection, and also exploring semantic shift along a spectrum of increasing distance, questions remain in my mind about some of the terminology, motivations, claims of novelty, and inconsistencies of reported results with existing numbers that need to be resolved.

---

> ### Comment · Reviewer_kPST · 2021-11-25
> **Post-rebuttal comments**
>
> Thanks for the response.
>
> I do not find the rebuttal about the applicability of the notion of concept shift convincing in the context of the submission. The definition in the so-called “conventional machine learning literature” is correct, but the term simply does not apply here. Take a look at the regression example in the cited paper for a case where it does apply appropriately. Attempting to (mis-)apply this “conventional” term ironically comes off as an attempt to “rebrand” current terminology. The provided example with shuffled pixels seems disingenuous — you strip semantics from structured data and then use unchanging semantics as an argument.
>
> Thanks for the discussion about empirical evidence about covariate shifts affecting norms in a minimizing fashion. It confirms my earlier comment — the development of the score is intuition and (limited-)empiricism based, and not particularly “theoretical”, as claimed in the submission. As the authors admit, covariate shift isn’t really disentangled from “concept” shift in the context of the submission, so the scores shouldn’t be assumed to be a valid disentanglement of the different types of shifts either.
>
> Thanks for the discussion about U vs. KL. I think KL numbers ought to be reported for all rows in all tables. If post-training calibration has been demonstrated to have a positive effect on OOD detection, that sounds like a good thing, since MSP is related to KL in some sense. So there might actually be a major advantage to using KL in that one might improve performance further by calibration, which you’d be unable to achieve with the proposed U score.
>
> Re. “their OOD detection methods do not attempt to address them differently”, Ahmed & Courville discuss that a model should not be rewarded for being sensitive to covariate shift. The desirable behaviour from a classifier, for example, is that it should be less sensitive to covariate shift and more sensitive to “semantic” shift. While it is certainly a safety-critical matter to be able to detect significant covariate shift when it happens because we anticipate classifiers to generalize poorly in such settings, the classifier itself should not be trained in a manner that promotes greater sensitivity to covariate shift, otherwise it is as if we want our deployed models to more biased.
>
> If [4]’s implementation reports weaker numbers, I still don’t see why we should use it as a baseline framework. I’d suggest using [9]’s implementations instead. The rebuttal says again that their method “does indeed achieve state of art results”, but the numbers are very often patently lower than existing methods like [9] (which hasn’t been included as a comparative baseline).
>
> In summary, I do not think the current version of the draft is quite ready for publication, and stand by my initial rating.

---

### Official Review · Reviewer_5ox7 · 2021-11-04

**Correctness:** 4
**Technical Novelty And Significance:** 3
**Empirical Novelty And Significance:** 3
**Recommendation:** 6
**Confidence:** 2

**Main Review:**

Strength:
1. This paper is well-written, and I enjoyed reading it. It is self-contained and also provides sufficient introduction to guide people less familiar with this topic.
2. The derivation of covariate shift score function and concept shift score function is solid and the justification/reasoning make a lot of sense.
3. Leveraging geometric sensitivity decomposition and model the learnable instance-dependent scalar alpha and beta is a neat idea. The design of neural network that approximates these two variables make sense, though more choices of activation can be studied in addition to the sigmoid and softplus.

Weakness/Question to Authors (order does not matter):
1. Overall, I like the explanation of Figure 1, but CIFAR 10 vs CIFAR 100 is a quite extreme example, because they are using the same source of data but categorized differently (I believe that CIFAR100 is a more fine-grained label set). Does concept shift always happens when P(X) follow the identical distribution? It seems not to be the case, given that SVHN simultaneously has concept and covariant shift. Also, I believe here you did not explain near OOD and far OOD? Is near OOD the case where either covariant or concept shift happens and far OOD the case where they both happen?
2. In section 3.4, is the reason why intra order-preserving function can better calibrate complex function in deep neural networks a conclusion of the theorem 1 in Rahimi et. Al. 2020? If so, probably it would be better to make this more explicit.


**Summary Of The Paper:**

This paper studies the problem of Out-Of-Distribution detection, which a focus on analyzing the covariate and concept shift. Leveraging these analysis, it further investigate score functions that capture sensitivity to those shifts and then theoretically derive new score functions, to improve OOD detection. This proposed method leads to a calibration function that obtain the state-of-the-art calibration performance on both in-distribution and out-of-distribution data. Experiments are conducted on small image recognition benchmarks, such as CIFAR100 and SVHN.


**Summary Of The Review:**

I am not an expert in OOD so my comments needs to be largely discounted. With that being said, I like this paper overall.  I believe the paper is very clearly written, well justified, and provides strong theoretical result and sufficient empirical result. Given this, I am leaning towards accepting this paper.

---

### Decision · Program_Chairs · 2022-01-20

**Decision:**

Reject

**Comment:**

This paper proposes a method for detecting two types of distributional shifts: covariate shifts in the input space  (due to input corruption) and semantic shifts (due to test data falling outside the support set of ID classes). The idea is based on the decomposition of KL-divergence between softmax prediction and a uniform vector. Furthermore, the authors propose Geometric ODIN to improve OOD detection and calibration, outperforming strong baseline on CIFAR10, CIFAR 100, and SVHN datasets. The paper aims to solve a very important problem in ML and the approach is thought-provoking.

However, there were several questions and confusions raised by the reviewers, such as the applicability of the model, justification of use of feature norm, discussion on sensitivity vs robustness, framing of the novelty, clear definition of OOD detection, definition of parameters, etc. (please see reviews for a comprehensive list). I invite authors to incorporate these points in the next version of the paper which will significantly improve the paper.